# COVID-19 infections and outcomes in a live registry of heart failure patients across an integrated health care system

César Caraballo[1,2☯], Megan McCullough[3☯], Michael A. Fuery[3☯], Fouad Chouairi[4☯], Craig Keating[5], Neal G. Ravindra[1], P. Elliott Miller[1], Maricar Malinis[6], Nitu Kashyap[5], Allen Hsiao[5], F. Perry Wilson[7], Jeptha P. Curtis[1,2], Matthew Grant[3], Eric J. Velazquez[1], Nihar R. Desai[1,2], Tariq Ahmad[1,2]*

1 Department of Internal Medicine, Section of Cardiovascular Medicine, Yale School of Medicine, New Haven, CT, United States of America, 2 Center for Outcomes Research & Evaluation (CORE), Yale New Haven Hospital, New Haven, CT, United States of America, 3 Department of Internal Medicine, Yale School of Medicine, New Haven, CT, United States of America, 4 Yale University School of Medicine, New Haven, CT, United States of America, 5 Joint Data Analytics Team, Yale New Haven Hospital, New Haven, CT, United States of America, 6 Department of Internal Medicine, Section of Infectious Diseases, Yale School of Medicine, New Haven, CT, United States of America, 7 Department of Internal Medicine, Section of Nephrology, Yale School of Medicine, New Haven, CT, United States of America

☯ These authors contributed equally to this work.
* tariq.ahmad@yale.edu

**Data Availability Statement:** Data are available from the Yale Institutional Data Access / Ethics Committee (contact via teshia.johnson@yale.edu)

## Abstract

### Background

Patients with comorbid conditions have a higher risk of mortality with SARS-CoV-2 (COVID-19) infection, but the impact on heart failure patients living near a disease hotspot is unknown. Therefore, we sought to characterize the prevalence and outcomes of COVID-19 in a live registry of heart failure patients across an integrated health care system in Connecticut.

### Methods

In this retrospective analysis, the Yale Heart Failure Registry (NCT04237701) that includes 26,703 patients with heart failure across a 6-hospital integrated health care system in Connecticut was queried on April 16th, 2020 for all patients tested for COVID-19. Sociodemographic and geospatial data as well as, clinical management, respiratory failure, and patient mortality were obtained via the real-time registry. Data on COVID-19 specific care was extracted by retrospective chart review.

### Results

COVID-19 testing was performed on 900 symptomatic patients, comprising 3.4% of the Yale Heart Failure Registry (N = 26,703). Overall, 206 (23%) were COVID-19+. As compared to COVID-19-, these patients were more likely to be older, black, have hypertension, coronary artery disease, and were less likely to be on renin angiotensin blockers (P<0.05, all). COVID-19- patients tended to be more diffusely spread across the state whereas

for researchers who meet the criteria for access to confidential data that could potentially lead to identification of patients.

**Funding:** The author(s) received no specific funding for this work.

**Competing interests:** The authors have declared that no competing interests exist.

COVID-19+ were largely clustered around urban centers. 20% of COVID-19+ patients died, and age was associated with increased risk of death [OR 1.92 95% CI (1.33–2.78); P<0.001]. Among COVID-19+ patients who were ≥85 years of age rates of hospitalization were 87%, rates of death 36%, and continuing hospitalization 62% at time of manuscript preparation.

## Conclusions

In this real-world snapshot of COVID-19 infection among a large cohort of heart failure patients, we found that a small proportion had undergone testing. Patients found to be COVID-19+ tended to be black with multiple comorbidities and clustered around lower socioeconomic status communities. Elderly COVID-19+ patients were very likely to be admitted to the hospital and experience high rates of mortality.

## Introduction

The novel coronavirus disease (COVID-19) caused by severe acute respiratory syndrome coronavirus-2 (SARS-CoV-2) has become a pandemic since its initial outbreak in December 2019 in Wuhan, China [1]. Risk factors for severe outcomes have been increasingly reported. Early reports from China indicated that patients with underlying cardiovascular disease had a 10.5% mortality rate compared with 2.3% case fatality rate in the general population [2]. Data from European and North American infection hotspots suggest that older age, hypertension, diabetes, and coronary artery disease are associated with higher odds of mortality [3, 4]. Therefore, it might be postulated that COVID-19 may be uniquely endangering to patients with heart failure, the most common cause of death and disability in the United States. However, very little is known about the prevalence of infection among heart failure patients in COVID-19 endemic regions and the risk factors associated with clinical deterioration.

The Yale New Haven Health System is one of the largest academic integrated health care systems in the U.S., its 6 hospitals (2,681 licensed beds) care for the majority of heart failure patients in Connecticut. We created a curated real-time registry within this system—the Yale Heart Failure Registry (NCT04237701)—that currently includes 26,703 patients with a diagnosis of HF and contains detailed information about demographics, comorbidities, laboratory and imaging studies, medications, and clinical outcomes including hospital readmissions. The aim of this report is to describe the demographic characteristics, coexisting conditions, and early outcomes among this real- world registry of heart failure patients tested for COVID-19 in a hotspot.

## Methods

### Study cohort

We queried the Yale Heart Failure Registry (NCT04237701) on April 16th, 2020 for patients who were tested for COVID-19. Briefly, the Yale Heart Failure Registry is a live registry of patients seen within the integrated Yale New Haven Health System (Fig 1) that meet the following evaluation logic: the patient has at least one encounter within the health system in the last 3 years and meets one of the four criteria (1) Problem List contains the Systematized Nomenclature of Medicine–Clinical Terms (SNOMED-CT) concept heart failure, (2) Encounter Diagnosis contains SNOMED-CT concept heart failure for two or more face to face

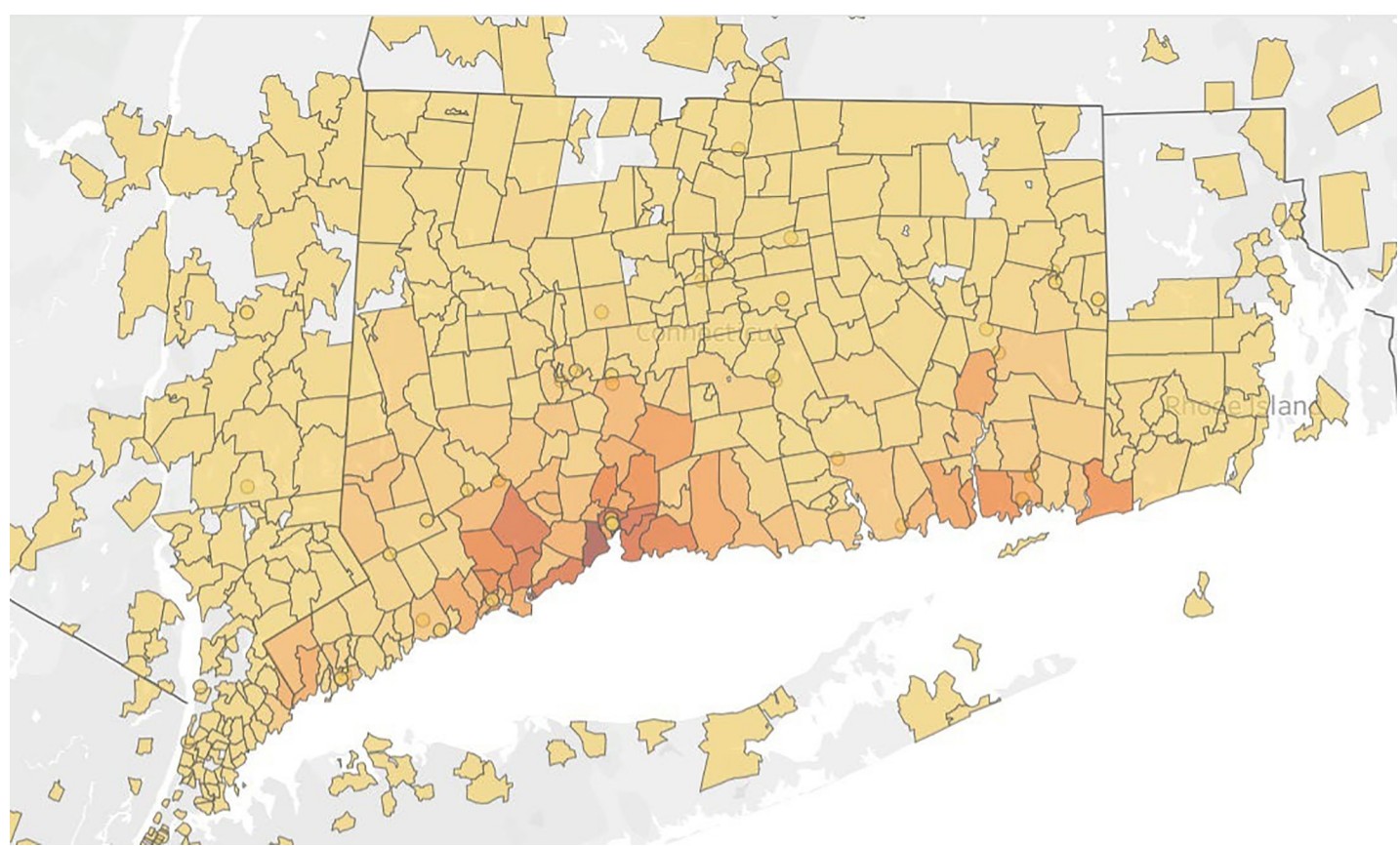

**Fig 1. Population distribution of patients in the Yale Heart Failure Registry**\*. \*Darker shading represents a greater density of patients in the registry from a zip code.

encounters within the last 12 months, (3) Encounter Diagnosis contains a SNOMED-CT concept heart failure for at least 1 admission within the last 12 months, (4) Coded Diagnosis is contained within one of the DRGs (291, 292, 293) for Heart Failure for at least 1 admission within the last 12 months. Of note, conditions in the Yale Electronic Health Record (EHR) use SNOMED-CT as the standard vocabulary for diagnosis codes. Registry exclusions are age<18, cardiac failure after obstetrical surgery or other procedure, or cardiac insufficiency during or resulting from a procedure. The registry has detailed information about demographics, comorbidities, laboratories, left ventricular ejection fraction (LVEF), medications, geospatial mapping, and clinical outcomes (including hospital readmissions). Registry data is linked to the National Death Index, a centralized database of death record information on file in state vital statistics offices. The registry is updated daily and can be queried for clinically relevant questions aimed at research and quality improvement. The registry has undergone extensive quality control by independent cardiologists and pharmacists; it has been found to have near identical patient characteristics to published registry data on heart failure patients [5]. The institutional review board (IRB) at the Yale University School of Medicine approved the study and waived the requirement for informed consent.

## Standardization of COVID-19 treatment across the health system

Yale New Haven Health System has one single integrated electronic health record system across 6 hospitals. This single instance of Epic (Epic Corporation, Verona WI) has a common

drug formulary, set of decision support tools, and order sets standardized across the System. Expert clinicians from different hospitals participated to create standardized COVID-19 order sets, documentation templates, and smart phrases for use in all hospitals. Physicians and other providers, however, can enter specific individual orders outside of COVID order sets as appropriate for individual patients but there is system wide guidance to standardize treatment (S1 and S2 Figs). In brief, admitted patients with confirmed SARS-CoV-2 infection are initiated on treatment if they were either hypoxic or had evidence of lower respiratory tract involvement (either by symptoms and/or chest radiography). Hydroxychloroquine is utilized for its antiviral properties. For patients who required mechanical ventilation, ECMO or had a combination of hypoxia with a hyperinflammatory state, tocilizumab is used for its anti- inflammatory properties via IL-6 receptor inhibition.

## Statistical analysis

Continuous variables are presented as median and interquartile range (IQR) and categorical variables are expressed as number of patients (percentage). χ2 and Mann- Whitney rank sum tests was used to compare categorical variables. Predictors of testing COVID-19+ was done using stepwise logistic regression with selection by Akaike information criterion (AIC) and evaluated by a C-statistic and DeLong's test (R version 3.5.3, MASS and pROC package). Logistic regression was used to assess the association between demographic variables and outcomes. All statistical tests were 2- tailed, and statistical significance was defined as $P<0.05$. Analyses were performed using Stata version 16.1 (Statacorp).

## Results

Based on data extracted from the Yale Heart Failure Registry on April 16th, 2020, we found that 900 patients, 3.4% of the total registry patients (N = 26,703) had been tested for SARS--COV-2. Per the Yale New Haven Health System protocol, testing was only performed on symptomatic patients. Of note, in the overall Yale Haven Health system, 22,254 patients had been tested and 6,357 (30.1%) were COVID-19+ as of April 16th2020. Most recent algorithms for treatment of COVID+ patients across the entire health care system are shown in S1 and S2 Figs. Patients in the registry, stratified by COVID-19 testing versus not and according to result, are shown in Table 1.

Of the HF registry patients tested for COVID-19, 23% were positive. Those who tested positive were significantly more likely to be older, black, hypertensive, and with coronary artery disease. They had higher BUN and creatinine at baseline. They were significantly less likely to be on renin angiotensin system (RAS) blockers at time of diagnosis. Of note, use of Sodium-glucose co-transporter-2 inhibitors (SGLT2i), Direct- Acting Oral Anticoagulants (DOACs), diuretics, and statins were also lower in this patient cohort. A significantly higher percentage of these patients were on Medicaid than in the overall cohort. Multivariable modeling revealed that black race, hypertension, lung disease, less DOAC and statin use, and number of prior hospital admissions were all independent predictors of testing positive for COVID-19 ($P<0.05$ all; AUC 0.69).

Fig 1 demonstrates the geospatial mapping of all patients who are included in the Yale Heart Failure Registry. It includes patients from all over Connecticut, Westchester County, NY, and Western Rhode Island. Fig 2 shows patients who tested positive and negative for COVID-19, with each dot representing a geographical location. As shown, patients who tested negative tended to more diffusely spread across the state, but those who tested positive were largely clustered around the cities of Bridgeport and New Haven, Connecticut.

**Table 1. Baseline clinical characteristics of patients according to testing status.**

| Characteristic | COVID-19 Not Tested | COVID-19 Tested | P | COVID-19⁻ | COVID-19⁺ | P |
|---|---|---|---|---|---|---|
| Number | 25,803 | 900 | | 694 | 206 | |
| Age | 76 (65–85) | 73 (62–82) | <0.001 | 72 (61–81) | 78 (65–87) | <0.001 |
| Female | 11,701 (45.3%) | 457 (50.8%) | <0.001 | 344 (49.6%) | 113 (54.9%) | 0.18 |
| Black | 3,414 (13.2%) | 205 (22.8%) | <0.001 | 143 (20.6%) | 62 (30.1%) | <0.05 |
| White | 19,824 (76.8%) | 598 (66.4%) | <0.001 | 479 (69.0%) | 119 (57.8%) | <0.05 |
| Medicare | 11,032 (42.8%) | 334 (37.1%) | <0.001 | 268 (38.6%) | 66 (32.0%) | 0.086 |
| Medicaid | 2,040 (7.9%) | 115 (12.8%) | <0.001 | 95 (13.7%) | 20 (9.7%) | 0.13 |
| BMI | 28.91 (24.82–34.41) | 29.12 (24.26–35.26) | 0.84 | 29.28 (24.43–35.26) | 28.67 (23.91–35.14) | 0.29 |
| HFrEF | 4,052 (22.8%) | 189 (21.0%) | 0.14 | 153 (22.0%) | 36 (17.5%) | 0.16 |
| Hypertension | 18,001 (69.8%) | 665 (73.9%) | 0.008 | 501 (72.2%) | 164 (79.6%) | <0.05 |
| COPD | 8,006 (31.0%) | 263 (29.2%) | 0.25 | 196 (28.2%) | 67 (32.5%) | 0.24 |
| CAD | 8,006 (31.0%) | 269 (29.9%) | 0.47 | 196 (28.2%) | 73 (35.4%) | <0.05 |
| Renal Disease | 7,131 (27.6%) | 317 (35.2%) | <0.001 | 238 (34.3%) | 79 (38.3%) | 0.28 |
| NT-proBNP | 1733 (592–4714) | 2092 (648–6515) | <0.001 | 2276 (704–6598) | 1357 (502–5281) | <0.05 |
| Sodium | 140 (138–141) | 139 (137–142) | 0.016 | 139 (137–142) | 139 (137–142) | 0.49 |
| Chloride | 103 (100–105) | 102 (98–105) | 0.003 | 102 (98–105) | 102 (99–106) | 0.68 |
| Creatinine | 1.1 (.88–1.46) | 1.2 (.88–1.9) | <0.001 | 1.19 (.87–1.88) | 1.3 (.93–1.97) | <0.05 |
| BUN | 22 (16–31) | 26 (17–41.5) | <0.001 | 25 (16–40) | 30 (19–46) | <0.05 |
| HbA1c | 6.2 (5.7–7.3) | 6.3 (5.7–7.5) | 0.08 | 6.3 (5.7–7.5) | 6.5 (5.7–7.7) | 0.6 |
| ACE-I/ARB | 13,354 (51.8%) | 312 (34.7%) | <0.001 | 254 (36.6%) | 58 (28.2%) | <0.05 |
| ARNI | 2,077 (8.0%) | 54 (6.0%) | 0.026 | 48 (6.9%) | 6 (2.9%) | <0.05 |
| Beta Blocker | 13,568 (52.6%) | 429 (47.7%) | 0.004 | 335 (48.3%) | 94 (45.6%) | 0.51 |
| CCB | 6,879 (26.7%) | 283 (31.4%) | 0.001 | 214 (30.8%) | 69 (33.5%) | 0.47 |
| SGLT2i | 810 (3.1%) | 25 (2.8%) | 0.54 | 24 (3.5%) | 1 (0.5%) | 0.02 |
| Warfarin | 2,811 (10.9%) | 81 (9.0%) | 0.07 | 65 (9.4%) | 16 (7.8%) | 0.48 |
| NOAC | 7,052 (27.3%) | 267 (29.7%) | 0.12 | 220 (31.7%) | 47 (22.8%) | 0.01 |
| Diuretic | 15,913 (61.7%) | 518 (57.6%) | 0.013 | 419 (60.4%) | 99 (48.1%) | 0.002 |
| Statin | 16,761 (65.0%) | 556 (61.8%) | 0.05 | 439 (63.3%) | 117 (56.8%) | <0.05 |

Data are presented as median (IQR) for continuous measures, and n (%) for categorical measures.

Testing for associations between baseline demographics (age, sex, race, BMI, and HFrEF) and key clinical outcomes: intensive care unit (ICU) admission, intubation, and death revealed the following: BMI was associated with an increased risk of ICU admission [OR 1.04 95% CI (1.00–1.08); $P$ = 0.036] and intubation [OR 1.07 95% CI (1.02–1.12); $P$ = 0.004]. Only age (increments of 10 years) was associated with increased risk of death [OR 1.92 95% CI (1.33–2.78); $P$<0.001].

Table 2 compares the clinical characteristics and admission laboratories of COVID-19 + patients who are alive versus who died at the time of this analysis. Of the heart failure patients testing positive for the virus, 20% died. These patients were more likely to be older (mean age 87) and white. Most patients who died had heart failure with preserved ejection fraction (HFpEF; ≈90%) and high rates of comorbid conditions. They also had significantly more abnormal values for baseline BUN, Creatinine, and N- terminal pro b-type natriuretic peptide (NT-proBNP), reflecting more advanced cardiovascular and renal disease. Additionally, they had significantly higher levels of COVID-19 specific laboratory test abnormalities including higher LDH, troponin T, procalcitonin, INR, Ferritin, CRP, AST, and absolute lymphocyte counts. Of the 206 patients who were COVID-19+, the majority (67%) were ≥85

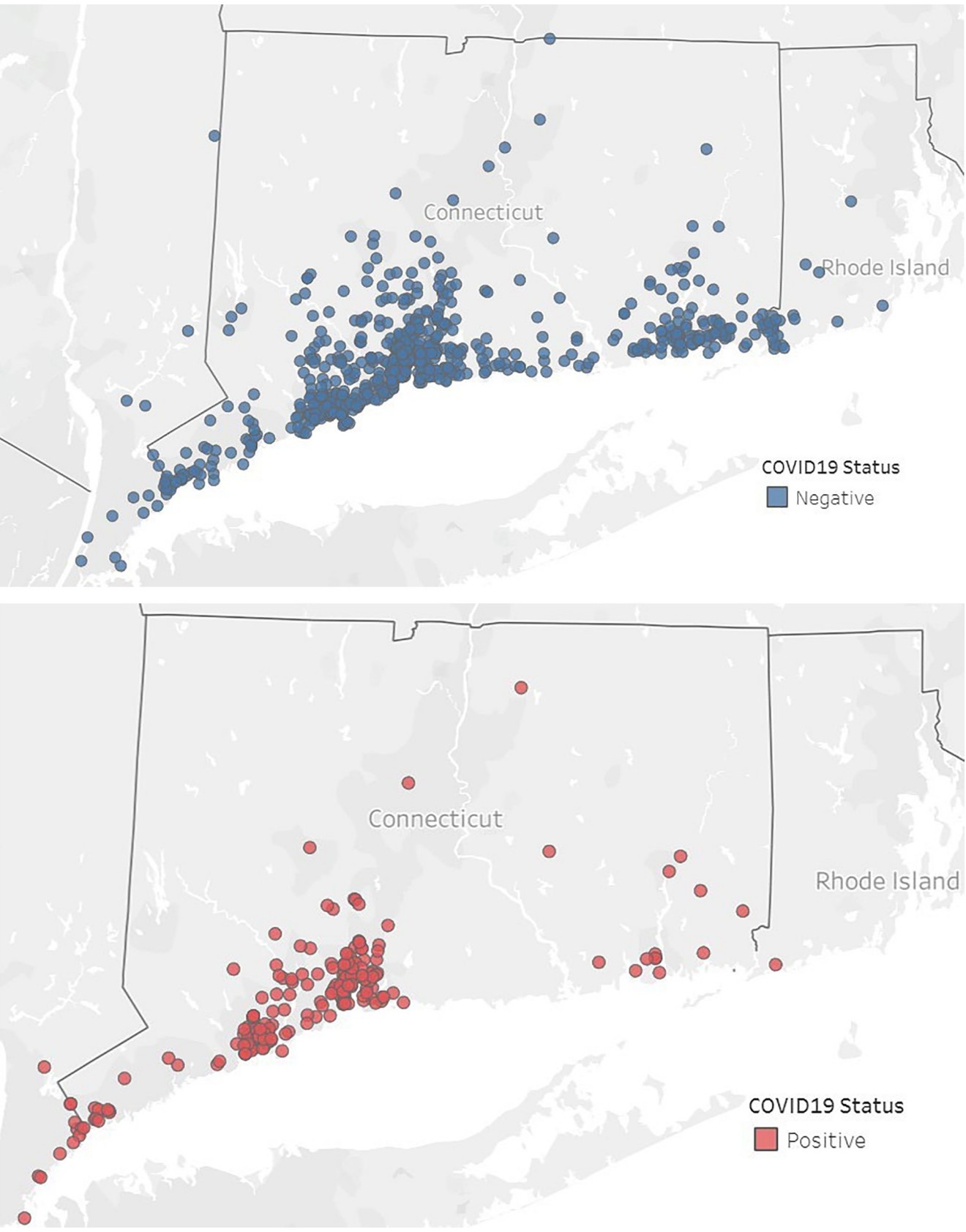

**Fig 2. (a).** Geospatial mapping of heart failure patients who tested negative for COVID-19. Dots represent individual patient home addresses. **(b).** Geospatial mapping of heart failure patients who tested positive for COVID-19. Dots represent individual patient home addresses.

years of age. In this cohort, rates of hospitalization were 87%. At the time of this analysis, 34 (62%) of these patients remained hospitalized and 20 (36%) had died.

Fig 3 shows medical and supportive interventions among COVID-19+ patients according to whether they are alive or dead at time of data extraction. Those who died were significantly more likely to have been treated with hydroxychloroquine, azithromycin, and atazanavir, likely reflecting more advanced disease. Those who died were also significantly more likely to be admitted to the ICU and require intensive oxygen support and intubation.

## Discussion

In this real-world analysis of a live heart failure registry across a large integrated health care system, we found that <4% of patients (900/26,703) had symptoms that met criteria and were

**Table 2. Characteristics of COVID-19+ patients according to clinical outcome.**

| Characteristic | COVID-19+ (Alive) | COVID-19+ (Dead) | P |
|---|---|---|---|
| Number | 172 | 34 | |
| Age | 76 (62–85) | 87 (80–89) | <0.001 |
| Female Sex | 96 (55.8%) | 17 (50.0%) | 0.53 |
| Black | 56 (32.6%) | 6 (17.6%) | 0.21 |
| White | 95 (55.2%) | 24 (70.6%) | |
| BMI | 28.67 (23.65–35.17) | 28.235 (24.64–33.21) | 0.98 |
| HFrEF | 33 (19.2%) | 3 (8.8%) | 0.15 |
| Hypertension | 138 (80.2%) | 26 (76.5%) | 0.62 |
| COPD | 55 (32.0%) | 12 (35.3%) | 0.71 |
| CAD | 55 (32.0%) | 18 (52.9%) | 0.02 |
| Kidney Disease | 60 (34.9%) | 19 (55.9%) | 0.02 |
| NT-proBNP | 1289 (435–4016) | 4275 (1090–7351) | 0.03 |
| Sodium | 139 (137–142) | 139 (135–142) | 0.48 |
| Chloride | 102 (99–106) | 102 (96–106) | 0.8 |
| Creatinine | 1.2 (0.9–1.8) | 1.7 (1.3–2.9) | <0.01 |
| BUN | 27.5 (18–45) | 37 (25–48) | 0.03 |
| HbA1c | 6.5 (5.7–7.6) | 6.4 (5.9–8.1) | 0.73 |
| ACE-I/ARB | 51 (29.7%) | 7 (20.6%) | 0.28 |
| Beta Blocker | 72 (41.9%) | 22 (64.7%) | 0.015 |
| LDH | 298 (242–408) | 373 (268–419) | 0.05 |
| Troponin T | 0.02 (0–0.05) | 0.06 (0.02–0.13) | <0.001 |
| Procalcitonin | 0.16 (0.08–0.41) | 0.37 (0.16–1.13) | 0.004 |
| INR | 1.05 (0.99–1.18) | 1.13 (1.03–1.42) | 0.04 |
| Fibrinogen | 500 (383–597) | 480 (371–611) | 0.95 |
| Ferritin | 501 (186–876) | 694 (233–1524) | 0.05 |
| D-dimer | 1.25 (0.73–2.44) | 1.19 (0.75–3.30) | 0.40 |
| CRP | 50 (13–119) | 85 (24–213) | 0.05 |
| AST | 34 (22–50) | 43 (30–87) | 0.01 |
| ALC | 0.9 (0.6–1.3) | 0.6 (0.4–0.9) | <0.01 |

Data are presented as median (IQR) for continuous measures, and n (%) for categorical measures.

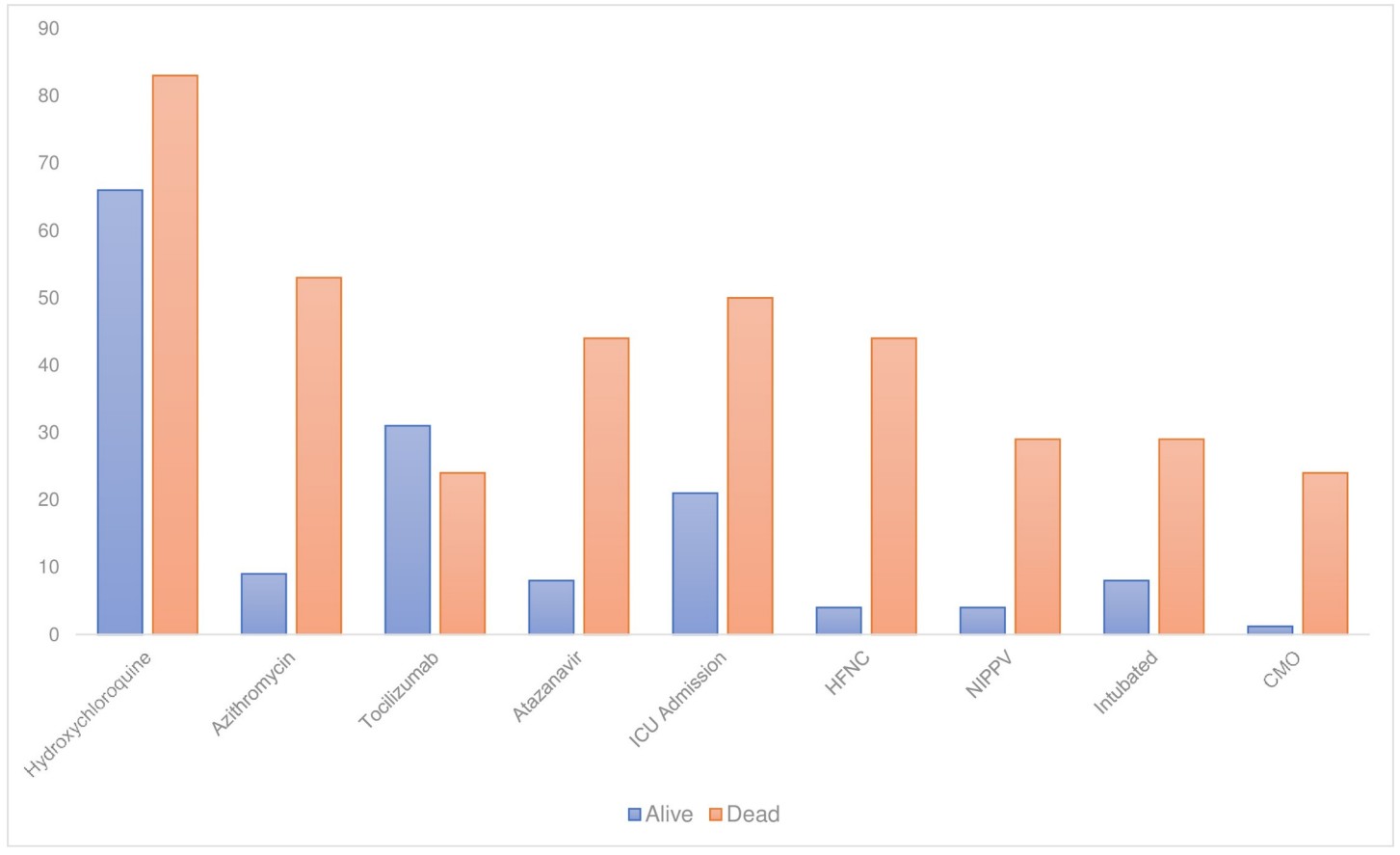

**Fig 3. Medical and supportive interventions in COVID-19+ patients according to clinical outcome.** ICU = Intensive Care Unit; HFNC = High flow nasal canula; NIPPV = Non-invasive positive pressure ventilation; CMO = comfort measures only.

able to get COVID-19 testing. Among the patients tested, 23% were positive, and they tended to cluster around the cities of Bridgeport and New Haven.

COVID-19+ patients were more likely to be black, have comorbid conditions, and were less likely to be on renin angiotensin blockade at time of testing. Of the COVID-19+ heart failure patients, 20% died, and they tended to be elderly (mean age 87) white patients with HFpEF. In their entirety, these data demonstrate the need for greater geographical based testing and interventions, particularly in vulnerable communities, and a greater focus on goals of care discussions among elderly heart failure patients who test positive.

This is the first report describing the impact of COVID-19 infection in a geographically contained population of heart failure patients under the purview of an integrated health care system, providing insight into how such a system can be used to analyze early data on diagnosis status and outcomes ascertainment among its population. We found that although testing was scattered across the state, those who tested positive tended to cluster around two cities—New Haven and Bridgeport, two of the cities with the lowest median household income in the state ($41,142 and $45,441 versus $76,106 for the state), and the highest percentage of black inhabitants (33.0% and 35.3%, versus 12% for the state) [6]. This disparity in both testing, and infection rates has been recently noted, with preliminary data showing that underrepresented minorities are developing COVID-19 infection more frequently [7]. The causes for this finding are multifaceted and have been hypothesized to include the disproportionate number of

minorities who are exposed to COVID-19 in their roles as "essential workers" and who spread the disease to their residentially segregated communities. Also, situations associated with lower socioeconomic conditions, such as housing insecurity and access to health care barriers, can facilitate the spread of infectious diseases. Taken together, these findings suggest an urgent need for a more nuanced strategy to tackle the disparate impact of the virus across vulnerable populations, including those with heart failure who are at particularly high risk.

Interestingly, we noted a significantly greater use of renin angiotensin blockade in both untested and negative patients in the registry. Compared to patients in the registry on an ACE-I/ARB, those not on these therapies were 50% more likely to be COVID-19+. Furthermore, rates were also significantly lower than in patients who tested negative.

Whereas these are cross-sectional observational data, they corroborate recent reports from China demonstrating a potential benefit of these therapies among COVID-19+ patients [8]. At the very least, this might suggest that these therapies do not increase risk in patients with heart failure and supports the AHA/ACC/HFSA statement supporting ongoing use of renin angiotensin blockade in heart failure patients during the pandemic [9].

Our study shows that a large portion of deaths occurred in elderly patients. High rates of mortality and morbidity have been described in older patients from Italy; however, less than 2% of their cohort was >80 years [3]. In our cohort, we noted 33% of COVID-19+ positive patients to be ≥85 years old. These patients had hospitalization rates of approximately 90%, 36% had died at the time of this analysis, and all the rest except one patient remained hospitalized. This suggests that special importance should be given to elderly individuals with HF when discussing testing distribution and early access to preventive therapies upon availability, such as vaccination against SARS-CoV-2. This also suggests an urgent need for early goals of care discussions in elderly heart failure patients who test positive for COVID-19+ given long hospitalizations and high mortality rates despite aggressive treatment [10, 11].

## Limitations

Our study has several notable limitations. First, we only included patients who were tested for COVID-19+ and other patients in the registry with the infection might have been uncounted due to either been asymptomatic, did not pursue testing for symptoms, or may have not been offered diagnostic testing. This is currently the practice at most medical centers across the United States. As such our findings should be considered a conservative estimate of the expanse of COVID19+ HF patients in the communities covered by the registry. Second, despite laboratory testing and interventions currently being performed in a protocolized manner across the Yale New Haven Health Care System, some patients might have been hospitalized prior to widespread application of this care pathway. Third, many of the patients described in the study remain hospitalized, leading to likely underreporting of total deaths from COVID-19+ in our cohort. Finally, the study has limitations inherent in any registry, where a high degree of granularity about heart failure is missing and all testing is solely performed at the clinician's discretion.

## Conclusions

In this real-world snapshot of COVID-19 infection among a large cohort of heart failure patients under the purview of an integrated health care system during peak infection time, we found that a very small percentage underwent testing. Patients found to be COVID-19+ tended to be black with multiple comorbidities and clustered around lower socioeconomic status communities in the State. Elderly COVID-19+ patients were very likely to be admitted to the hospital and experience high rates of mortality.

## Supporting information

**S1 Fig. YNHHS initial treatment algorithm for hospitalized adults with non-severe COVID–19.**
(JPG)

**S2 Fig. YNHHS initial treatment algorithm for hospitalized adults with severe COVID–19.**
(JPG)

## Acknowledgments

We would like to thank the YNHH/YSM Ad-Hoc COVID-19 Treatment team for allowing us to share the Yale Treatment Protocol and the entire YNHH/YSM Healthcare team for their efforts is support of patients during this pandemic. We would also like to express immense gratitude to Mrs. Elizabeth Tobin and Mr. Gregory Kopchinsky for their generous support for the Yale Heart Failure Research Group.

## Author Contributions

**Conceptualization:** Megan McCullough, Michael A. Fuery, Nihar R. Desai, Tariq Ahmad.

**Data curation:** César Caraballo, Megan McCullough, Michael A. Fuery, Fouad Chouairi, Craig Keating, Nitu Kashyap, Allen Hsiao, Nihar R. Desai, Tariq Ahmad.

**Formal analysis:** César Caraballo, Megan McCullough, Michael A. Fuery, Craig Keating, P. Elliott Miller, Maricar Malinis, Jeptha P. Curtis, Eric J. Velazquez, Nihar R. Desai, Tariq Ahmad.

**Funding acquisition:** César Caraballo, Craig Keating, Neal G. Ravindra, F. Perry Wilson, Eric J. Velazquez, Tariq Ahmad.

**Investigation:** César Caraballo, Megan McCullough, Neal G. Ravindra, Maricar Malinis, F. Perry Wilson, Matthew Grant, Eric J. Velazquez, Tariq Ahmad.

**Methodology:** César Caraballo, Megan McCullough, Neal G. Ravindra, F. Perry Wilson, Matthew Grant, Tariq Ahmad.

**Project administration:** César Caraballo, Megan McCullough.

**Resources:** Craig Keating, Neal G. Ravindra, F. Perry Wilson, Matthew Grant, Tariq Ahmad.

**Software:** Maricar Malinis, Tariq Ahmad.

**Supervision:** Craig Keating, Tariq Ahmad.

**Validation:** Tariq Ahmad.

**Writing – original draft:** Tariq Ahmad.

**Writing – review & editing:** Tariq Ahmad.

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
