## [Decision Letter · Decision Letter 0]

18 Aug 2020

PONE-D-20-13485

COVID-19 Infections and Outcomes in a Live Registry of Heart Failure Patients Across an Integrated Health Care System

PLOS ONE

Dear Dr. Ahmad,

Thank you for submitting your manuscript to PLOS ONE. After careful consideration, we feel that it has merit but does not fully meet PLOS ONE’s publication criteria as it currently stands. Therefore, we invite you to submit a revised version of the manuscript that addresses the points raised during the review process.

We look forward to receiving your revised manuscript.

Kind regards,

Chiara Lazzeri

Academic Editor

PLOS ONE

Journal Requirements:

2. In ethics statement in the manuscript and in the online submission form, please provide additional information about the database used in your retrospective study. Specifically, please ensure that you have discussed whether all data were fully anonymized before you accessed them and/or whether the IRB or ethics committee waived the requirement for informed consent. If patients provided informed written consent to have their data used in research, please include this information.

4. We note that Figures 1 and 2 in your submission contain map images which may be copyrighted. All PLOS content is published under the Creative Commons Attribution License (CC BY 4.0), which means that the manuscript, images, and Supporting Information files will be freely available online, and any third party is permitted to access, download, copy, distribute, and use these materials in any way, even commercially, with proper attribution. For these reasons, we cannot publish previously copyrighted maps or satellite images created using proprietary data, such as Google software (Google Maps, Street View, and Earth). For more information, see our copyright guidelines: http://journals.plos.org/plosone/s/licenses-and-copyright.

4.1.    You may seek permission from the original copyright holder of Figures 1 and 2 to publish the content specifically under the CC BY 4.0 license. 

4.2.    If you are unable to obtain permission from the original copyright holder to publish these figures under the CC BY 4.0 license or if the copyright holder’s requirements are incompatible with the CC BY 4.0 license, please either i) remove the figure or ii) supply a replacement figure that complies with the CC BY 4.0 license. Please check copyright information on all replacement figures and update the figure caption with source information. If applicable, please specify in the figure caption text when a figure is similar but not identical to the original image and is therefore for illustrative purposes only.

Reviewers' comments:

Reviewer's Responses to Questions

**Comments to the Author**

1. Is the manuscript technically sound, and do the data support the conclusions?

Reviewer #1: Yes

Reviewer #2: Yes

2. Has the statistical analysis been performed appropriately and rigorously? 

Reviewer #1: Yes

Reviewer #2: Yes

3. Have the authors made all data underlying the findings in their manuscript fully available?

Reviewer #1: Yes

Reviewer #2: Yes

4. Is the manuscript presented in an intelligible fashion and written in standard English?

Reviewer #1: Yes

Reviewer #2: Yes

5. Review Comments to the Author

Reviewer #1: Caraballo C. and Collegues described the social and clinical profile, the prevalence and the predictors of outcomes of a large sample of HF patients positive to SARS-CoV-2, registered in the YALE Heart Failure Registry in Connecticut area.

Being able to use a real-time registry the authors have clearly described the clinical differences of 900 symptomatic COVID+ HF patients in respect those COVID- and the predictors of poor outcomes in this subgroup of HF patients.

Although some clinical data had already been confirmed elsewhere, in this study some interesting findings must be underlined: the elevated rate of death (20%) in very elderly (>85 years) HF patients with symptomatic COVID, the negative prognostic role of high burden of comorbid conditions as well as the low level of socioeconomic status. This data, together of other clinical evidences, clearly suggest how HF older frail patients should be the priority for any prevention program of vaccination or other preventive care program. Also interesting is the data concerning the positive association RAAS inhibitor drugs and prognosis; moreover this result is found in a large population of patients in whom these drugs have a high level of evidence based recommendation. Another flag finding concerns the low percentage of HF patients underwent testing for COVID, this report, in my opinion, is another important clinical alert that must be taken into account by those who have health- care system responsibilities.

Thus I think that this paper, well written, deserves a good consideration for publication in the Journal accounting the aforementioned considerations.

Obviously this is a retrospectively analysis but in my opinion it does not reduce significantly the clinical value.

Reviewer #2: The authors assessed how many patients include in their Yale Heart Failure Registry (N=26,703). Were tested for COVID-19 and were positive . they then compare the clinical characteristics of the positive patients versus the negative ones and show that covid positive patients tend to be more geographically concentrated, an expected finding based on the characteristic of the disease. Elderly COVID-19+ patients were very likely to be admitted to the hospital and experience high rates of mortality.

The study is clearly written. However, its clinical implications and novelty compared with current literature seem extremely limited. In the discussion, the authors state “This is the first report describing the impact of COVID-19 infection in a geographically contained population of heart failure patients under the purview of an integrated health care system.” I agree but is this worthwhile? Then, they state “We found that (…) those who tested positive tended to cluster around two cities—New Haven and Bridgeport, two of the cities with the lowest median household income in the state…” Again, this conforms data with most of infective disease.

6. PLOS authors have the option to publish the peer review history of their article (what does this mean?). If published, this will include your full peer review and any attached files.

Reviewer #1: No

Reviewer #2: No

---

## [Author Response · Author response to Decision Letter 0]

20 Aug 2020

Response to Reviewer Comments

Caraballo et al. PLOS ONE.

RE: PONE-D-20-13485

REVIEWER #1 

Caraballo C. and Colleagues described the social and clinical profile, the prevalence and the predictors of outcomes of a large sample of HF patients positive to SARS-CoV-2, registered in the YALE Heart Failure Registry in Connecticut area.

Being able to use a real-time registry the authors have clearly described the clinical differences of 900 symptomatic COVID+ HF patients in respect those COVID- and the predictors of poor outcomes in this subgroup of HF patients.

Although some clinical data had already been confirmed elsewhere, in this study some interesting findings must be underlined: the elevated rate of death (20%) in very elderly (>85 years) HF patients with symptomatic COVID, the negative prognostic role of high burden of comorbid conditions as well as the low level of socioeconomic status. This data, together of other clinical evidences, clearly suggest how HF older frail patients should be the priority for any prevention program of vaccination or other preventive care program. Also interesting is the data concerning the positive association RAAS inhibitor drugs and prognosis; moreover this result is found in a large population of patients in whom these drugs have a high level of evidence based recommendation. Another flag finding concerns the low percentage of HF patients underwent testing for COVID, this report, in my opinion, is another important clinical alert that must be taken into account by those who have health- care system responsibilities.

Thus I think that this paper, well written, deserves a good consideration for publication in the Journal accounting the aforementioned considerations.

Obviously this is a retrospectively analysis but in my opinion it does not reduce significantly the clinical value.

Response: We thank the reviewer for their kind and important comments. We have incorporated them into our manuscript.

We agree with the reviewer of the uttermost importance of increasing access to testing and vaccination to elderly patients with HF. We have modified the discussion section as shown below (additions underlined):

Pages 13 and 14, lines 217-224:

“In our cohort, we noted 33% of COVID-19+ positive patients to be ≥85 years old. These patients had hospitalization rates of approximately 90%, 36% had died at the time of this analysis, and all the rest except one patient remained hospitalized. This suggests that special importance should be given to elderly individuals with HF when discussing testing distribution and early access to preventive therapies upon availability, such as vaccination against SARS-CoV-2. This also suggests an urgent need for early goals of care discussions in elderly heart failure patients who test positive for COVID-19+ given long hospitalizations and high mortality rates despite aggressive treatment.”

REVIEWER #2

The authors assessed how many patients include in their Yale Heart Failure Registry (N=26,703). Were tested for COVID-19 and were positive . they then compare the clinical characteristics of the positive patients versus the negative ones and show that covid positive patients tend to be more geographically concentrated, an expected finding based on the characteristic of the disease. Elderly COVID-19+ patients were very likely to be admitted to the hospital and experience high rates of mortality.

The study is clearly written. However, its clinical implications and novelty compared with current literature seem extremely limited. In the discussion, the authors state “This is the first report describing the impact of COVID-19 infection in a geographically contained population of heart failure patients under the purview of an integrated health care system.” I agree but is this worthwhile? Then, they state “We found that (…) those who tested positive tended to cluster around two cities—New Haven and Bridgeport, two of the cities with the lowest median household income in the state…” Again, this conforms data with most of infective disease.

Response: We thank the reviewer for their comments, which have helped us improve our manuscript. Although everyday new data on COVID-19 is being published, its implications in patients with heart failure are still being elucidated and our study contributes to this knowledge by demonstrating early outcomes among these patients, using a health care system that facilitates outcomes detection and ascertainment in our region. We have modified the discussion to better reflect this, as shown below (additions underlined):

Page 12, lines 181-183:

“This is the first report describing the impact of COVID-19 infection in a geographically contained population of heart failure patients under the purview of an integrated health care system, providing insight into how such a system can be used to analyze early data on diagnosis status and outcomes ascertainment among its population.”

Also, we agree that the SARS-CoV-2 transmission among individuals with low SES may be a function of higher spread of infectious diseases in general. We have added this as one of the plausible explanations of our findings (additions underlined):

Page 13, lines 196-198:

“The causes for this finding are multifaceted and have been hypothesized to include the disproportionate number of minorities who are exposed to COVID-19 in their roles as “essential workers” and who spread the disease to their residentially segregated communities. Also, situations associated with lower socioeconomic conditions, such as housing insecurity and access to health care barriers, can facilitate the spread of infectious diseases. Taken together, these findings suggest an urgent need for a more nuanced strategy to tackle the disparate impact of the virus across vulnerable populations, including those with heart failure who are at particularly high risk.”

---

## [Editor Report · Decision Letter 1]

26 Aug 2020

COVID-19 Infections and Outcomes in a Live Registry of Heart Failure Patients Across an Integrated Health Care System

PONE-D-20-13485R1

Dear Dr. Ahmad,

We’re pleased to inform you that your manuscript has been judged scientifically suitable for publication and will be formally accepted for publication once it meets all outstanding technical requirements.

Kind regards,

Chiara Lazzeri

Academic Editor

PLOS ONE
---

## [Editor Report · Acceptance letter]

4 Sep 2020

PONE-D-20-13485R1 

COVID-19 Infections and Outcomes in a Live Registry of Heart Failure Patients Across an Integrated Health Care System 

Dear Dr. Ahmad:

I'm pleased to inform you that your manuscript has been deemed suitable for publication in PLOS ONE. Congratulations! Your manuscript is now with our production department. 

Kind regards, 

on behalf of

Dr. Chiara Lazzeri 

Academic Editor

PLOS ONE